# Symptom-Related Distress among Indigenous Australians in Specialist End-of-Life Care: Findings from the Multi-Jurisdictional Palliative Care Outcomes Collaboration Data

**DOI:** 10.3390/ijerph17093131

**Published:** 2020-04-30

**Authors:** John A. Woods, Claire E. Johnson, Hanh T. Ngo, Judith M. Katzenellenbogen, Kevin Murray, Sandra C. Thompson

**Affiliations:** 1Western Australian Centre for Rural Health, School of Population and Global Health, The University of Western Australia, Perth, WA 6009, Australia; sandra.thompson@uwa.edu.au; 2Eastern Health, Melbourne, VIC 3128, Australia; Claire.Johnson@easternhealth.org.au; 3School of Nursing and Midwifery, Monash University, Melbourne, VIC 3800, Australia; 4Discipline of Emergency Medicine, Medical School, The University of Western Australia, Perth, WA 6009, Australia; hanh.ngo@rcswa.edu.au; 5Rural Clinical School of Western Australia, The University of Western Australia, Perth, WA 6009, Australia; 6School of Population and Global Health, The University of Western Australia, Perth, WA 6009, Australia; judith.katzenellenbogen@uwa.edu.au (J.M.K.); kevin.murray@uwa.edu.au (K.M.); 7Telethon Kids Institute, Northern Entrance, Perth Children’s Hospital, Perth, WA 6009, Australia

**Keywords:** Aboriginal, palliative care, terminal care, symptom assessment, pain, psychological distress

## Abstract

Symptom relief is fundamental to palliative care. Aboriginal and Torres Strait Islander (Indigenous) Australians are known to experience inequities in health care delivery and outcomes, but large-scale studies of end-of-life symptoms in this population are lacking. We compared symptom-related distress among Indigenous and non-Indigenous Australian patients in specialist palliative care using the multi-jurisdictional Palliative Care Outcomes Collaboration dataset. Based on patient-reported rating scale responses, adjusted relative risks (aRRs) stratified by care setting were calculated for occurrence of (i) symptom-related moderate-to-severe distress and worsening distress during a first episode of care and (ii) symptom-related moderate-to-severe distress at the final pre-death assessment. The *p*-value significance threshold was corrected for multiple comparisons. First-episode frequencies of symptom-related distress were similar among Indigenous (*n* = 1180) and non-Indigenous (*n* = 107,952) patients in both inpatient and community settings. In final pre-death assessments (681 Indigenous and 67,339 non-Indigenous patients), both groups had similar occurrence of moderate-to-severe distress when care was provided in hospital. In community settings, Indigenous compared with non-Indigenous patients had lower pre-death risks of moderate-to-severe distress from overall symptom occurrence (aRR 0.78; *p* = 0.001; confidence interval [CI] 0.67–0.91). These findings provide reassurance of reasonable equivalence of end-of-life outcomes for Indigenous patients who have been accepted for specialist palliative care.

## 1. Introduction

People with a life-limiting illness often experience physical symptoms that distress them and their carers as well as treating clinicians [1,2]. Clusters of multiple concurrent symptoms occur commonly among patients with advanced cancer [3] or other life-limiting illnesses [4], and have a profound impact on quality of life [5]. Relief of such symptoms is a core function of palliative care [6] and is especially important when death is imminent [7]. Compared with other Australians, Aboriginal and Torres Strait Islander (hereafter respectfully referred to as Indigenous) people experience cultural and other barriers to high-quality health care [8,9] along with major disparities in health outcomes [10,11]. Impaired communication between Indigenous patients and clinicians [12,13,14] potentially constrains effective identification and management of symptoms. Although the experiences of Indigenous Australians in palliative care, including perspectives on symptom control [15], have been explored qualitatively, there have been no large-scale studies of end-of-life symptoms among Indigenous patients.

The Australian Palliative Care Outcomes Collaboration (PCOC) undertakes routine point-of-care data collection from and regular reporting back to participating end-of-life care services nationwide, for the purposes of independent quality appraisal and benchmarking of outcomes [16]. At the time the data for this study were collected, participation in PCOC was restricted to specialist palliative care services. The hierarchically nested longitudinal PCOC dataset comprises (i) patient-level details recorded at entry to care by a participating service, (ii) data on each episode of care provided to the patient, and (iii) clinical assessment data routinely recorded at one or more points within each episode [17].

Using PCOC data, we previously compared the clinical status at entry to palliative care of Indigenous compared with non-Indigenous Australian patients experiencing a life-limiting illness [18]. We herein have complemented these findings by investigating symptom-related distress among Indigenous compared with non-Indigenous patients after a care episode has commenced and as death approaches. Our first objective was to detect differences between the two patient groups in the burden of distress attributable to one or several symptoms, rated by the patient following initiation of an episode of care, as well as the worsening of symptom-related distress during an episode compared with the level at episode commencement. Our second objective was to compare the two patient groups in relation to self-reported symptom-related distress at the point of the final assessment prior to death.

## 2. Materials and Methods

### 2.1. Study Design and Data Source

This retrospective cohort study was based on multi-jurisdictional PCOC data collected during the period from 1 January 2010 to 30 June 2015, with episode of care as the unit of observation. Routine point-of-care standardised clinical assessments based on validated instruments are undertaken by participating services at the commencement of each care episode, then at the point of transition between each clinical ‘phase’ during the episode, and also at the end of an episode for surviving patients but not for those whose episode has ended in death [17]. ‘Phase’ of illness is a validated indicator of a period in a patient’s condition defined in relation to changing care needs, and is categorised non-sequentially as stable, unstable, deteriorating or terminal [19,20]. This study excluded ‘bereavement’ phases, which are captured in PCOC data to record care of a bereaved family following a patient’s death.

For the first study objective, the analysis was based on post-commencement symptom assessments, restricted for the purpose of between-patient comparability to first episodes of care by a participating service, which account for the majority (69.7%) of episodes captured in the dataset. For the second objective, we analysed data from the final assessment prior to death among the subgroup of patients with a recorded death during an episode of care. As an episode could be both the first provided to a patient and also result in death, the datasets used for the two analyses were overlapping.

### 2.2. Study Participants and Episode of Care Inclusion Criteria

Patients eligible for the study (Figure 1) were those aged ≥18 years at entry to care who had (i) a first episode of care in which symptom scores were recorded at least once after episode commencement, and/or (ii) a ‘pre-death’ episode (i.e., an episode that culminated in a recorded death) with a recorded pre-death assessment (i.e., documented commencement of a final phase) within seven days of death. First episodes of care provided to a patient by a service were identified by means of a pre-2010 look-back to dataset inception, undertaken by the PCOC data manager at the request of the researchers.

### 2.3. Study Measures

Patient details for the study (ascertained at commencement of care by a service) were the Indigenous identifier, sex, and the principal diagnosis. The Indigenous identification of a patient is recorded in PCOC data as Aboriginal, Torres Strait Islander, both Aboriginal and Torres Strait Islander, or neither. For the current study, this identifier was categorised as Indigenous (i.e., Aboriginal and/or Torres Strait Islander), non-Indigenous, or missing. The principal diagnosis, which is categorised in PCOC data in binary form (cancer or other), then further subdivided by body system into twenty-nine categories [17], was analysed as a 29-category nominal variable. Episode-level data were start date, patient age in years at commencement, and the setting of care (inpatient, hospital outpatient/day-case, or community). Symptom-related distress was measured by the Symptom Assessment Scale (SAS) [21]. This validated Likert-type instrument provides ratings by the patient (or occasionally by the carer, should the patient be incapacitated) of symptom-induced distress (from 0 [symptom absent] to 10 [most severe]) from the previous 24 h across seven symptom-specific items (pain, breathing problems, nausea, bowel problems, appetite problems, insomnia and fatigue). Responses for each item were dichotomised using the threshold corresponding with PCOC benchmarking practice [17]: absent-to-mild (<4) versus moderate-to-severe (4–10).

For Objective 1, the following outcomes following commencement of a first episode were investigated: (i) the occurrence of at least one instance of moderate-to-severe distress reported for any symptom; (ii) the occurrence of at least one instance of moderate-to-severe distress reported for each of at least three different symptoms; (iii) for each of the seven individual symptoms assessed in the SAS scale, the occurrence of at least one instance of moderate-to-severe distress; (iv) at least one instance of worsening distress (i.e., an increment of ≥2 SAS points [22]) compared with assessment at episode start) for any symptom; (v) at least one instance of worsening distress (compared with assessment at episode start) for at least three symptoms; and (vi) for each of the seven individual symptoms, at least one instance of worsening distress (compared with assessment at episode start).

For Objective 2, relative risks were modelled at the final assessment prior to death for (i) the occurrence of moderate-to-severe distress related to any symptom, (ii) the occurrence of moderate-to-severe distress related to at least three symptoms, and (iii) for each of the seven individual symptoms, the occurrence of moderate-to-severe distress.

### 2.4. Statistical Analysis

All outcomes were modelled as binary variables. Comparisons between Indigenous and non-Indigenous patients were calculated as crude and adjusted relative risks using Poisson regression with robust variance structure [20]. Age, sex, and principal diagnosis were the covariates included in all adjusted analyses. For Objective 2, an additional model included the varying time interval (in days) between the final assessment and death as an additional covariate. For both objectives, analyses were stratified by setting of care, except that subgroup analysis restricted to the outpatient/day admission setting was not performed because small numbers of Indigenous patients prevented model convergence. *P*-values of <0.05 were considered significant, with the false discovery rate corrected for multiple comparisons using the Benjamini–Hochberg method [23].

All analyses were restricted to records with non-missing values for all critical variables, and were conducted using Stata version 15.1 (Stata Corporation, College Station, TX, USA).

### 2.5. Ethics

The study was approved by the Western Australian Aboriginal Health Ethics Committee (project reference: #616, approval 17 September 2015; manuscript-specific approval 14 October 2019) and the University of Western Australia Human Research Ethics Committee (reference: RA/4/1/7441, 8 July 2016). The bodies that granted permission determined that individual consent of subjects was not required; data were analyzed anonymously and only aggregated data are presented.

## 3. Results

Among 144,536 adult patients with records in the dataset, 133,307 had sufficiently complete data for adjusted analyses, i.e., 7.8% of patients were excluded because of missing values across one or more critical variables (Figure 1). Of the patients excluded, 5.0% identified as Indigenous and 4.5% identified as non-Indigenous. Among patients with non-missing critical person-level data, records from the first episode of care were available for 128,998 (96.8% of total: Indigenous patients 96.7%; non-Indigenous patients 96.8%). Of these episodes, 109,132 (84.6%: Indigenous patients 85.6%; non-Indigenous patients 84.6%) resulted in survival through at least one phase and therefore had at least one reported post-commencement SAS assessment. Records of an episode ending in death were available for 79,554 patients (59.7%) with non-missing person-level values (Indigenous patients 55.6%; non-Indigenous patients 59.7%). Among 79,369 (99.8%: Indigenous 99.7%; non-Indigenous 99.8%) with an identifiable record corresponding with the phase culminating in death, 68,020 (85.7%: 86.0% of Indigenous and 85.7% of non-Indigenous patients) had such a phase with a duration of ≤7 days, and were included in the analysis. In total, 126,063 patients (1360 Indigenous; 124,703 non-Indigenous) were included in the study, with an overlap of 51,089 patients (40.5% of the total: 501 Indigenous; 50,588 non-Indigenous) who provided data for both first and pre-death episode analyses.

In both of the substantially overlapping groups of first and pre-death episodes included in the analyses, Indigenous patients were about a decade younger on average than non-Indigenous patients at episode start (Table 1). Across first and pre-death episodes, a slender majority of Indigenous patients were female, whereas males predominated slightly among the non-Indigenous patients. Cancer was the principal diagnostic category for 83.5% of first and 77.5% of pre-death episodes, with little difference between Indigenous and non-Indigenous patients in either episode grouping in the proportions of cancer diagnoses.

### 3.1. Objective 1: First Episodes

Following commencement of a first episode of care, symptom-related distress rated as moderate-to-severe (SAS score ≥4) occurred at least once in 78.3% of patients (Indigenous 77.8%; non-Indigenous 78.3%). Post-commencement moderate-to-severe ratings across at least three symptoms were reported in 42.1% of episodes (Indigenous 40.5%; non-Indigenous 42.2%). Fatigue was the symptom most frequently reported as causing high levels of moderate-to-severe distress (64.5%) and nausea the least commonly reported (13.8%). Worsening of symptom-related distress (increment ≥2 in any SAS score following commencement) was reported in 57.0% of episodes (Indigenous 54.4%; non-Indigenous 57.1%), with worsening of distress across three or more symptoms in 23.6% episodes (Indigenous 22.9%; non-Indigenous 23.6%). Fatigue (28.7%: Indigenous 26.9%; non-Indigenous 28.7%), pain (24.7%: Indigenous 24.0%; non-Indigenous 24.7%) and appetite problems (21.7%: Indigenous 20.4%; non-Indigenous 21.7%) were the symptoms most commonly reported as causing worsening distress.

In crude and adjusted estimates from the total sample and from inpatient and community care settings (Table 2), after correction for multiple comparisons there were no significant differences between Indigenous and non-Indigenous patients in the occurrence of at least one instance of moderate-to-severe distress due to any symptom, due to at least one worsening symptom, or due to moderate-to-severe distress from combined occurrence of at least three symptoms. Indigenous patients did not have a significantly higher risk of reporting distress due to the post-commencement occurrence or worsening of any of each of the symptoms considered individually, overall, or in either the inpatient or community setting. Across symptoms investigated individually in all settings combined, Indigenous compared with non-Indigenous patients had a significantly lower risk of reporting the occurrence of moderate-to-severe distress due to nausea and appetite problems but not in relation to the other symptoms investigated. No significant differences between the two patient groups in post-commencement occurrence of individual symptoms were evident in the inpatient setting. In the community setting, the lower risk of reporting symptom-related distress among Indigenous patients was significant only for appetite problems and fatigue. In relation to post-commencement worsening of individual symptoms, the only significant difference between the two patient groups was a lower risk among Indigenous patients of nausea-related distress in all settings combined.

### 3.2. Objective 2: ‘Pre-death’ Episodes

Among the 68,020 patients (Indigenous 681; non-Indigenous 67,339) included in the analysis of pre-death episodes, the average time interval between final assessment of clinical status and death (i.e., the length of the final phase) was 2.1 days (median 1 day; interquartile range 1–3 days) among both Indigenous and non-Indigenous patients. At the point of final assessment, moderate-to-severe distress related to at least one symptom was reported by 63.1% of patients, and at least three different symptoms by 24.4%. Among individual symptoms, moderate-to-severe distress was most commonly reported in relation to fatigue (44.9%), breathing problems (28.0%), and pain (26.8%).

At the final assessment prior to death, Indigenous compared with non-Indigenous patients did not report significantly higher levels of distress related to any symptom, to at least three symptoms, or to any of each of the individual symptoms analysed separately, overall or in either the inpatient or community setting (Table 3). Indigenous compared with non-Indigenous patients in the community setting but not in the inpatient setting or all settings combined had a lower adjusted risk of reporting moderate-to-severe distress due to any symptom. There were no differences between the two patient groups in reported distress due to combinations of three or more symptoms. Among individual symptoms, the only significant difference between the groups was a lower risk in the community setting of reporting moderate-to-severe distress due to appetite problems among Indigenous compared with non-Indigenous patients.

## 4. Discussion

We adapted previously published approaches to the investigation of symptom-related distress using PCOC data [22,24] to produce what we believe is the first multi-jurisdictional study addressing symptom control for Australian Indigenous patients receiving specialist end-of-life care. The study provides no evidence that symptom-related distress in this context is more frequent among Indigenous than non-Indigenous patients. By some indices, Indigenous patients cared for in community settings appear to have a lower risk than non-Indigenous patients of distress arising from uncontrolled symptoms at the point of final assessment prior to death.

However, a majority of both Indigenous and non-Indigenous patients overall, and in both inpatient and community settings, report at least one symptom causing moderate-to-severe distress and/or worsening symptom-related distress after commencement of a first episode of care. Prior to death, a majority of patients report at least one symptom of moderate-to-severe distress but only a minority report a combination of three or more symptoms causing moderate-to-severe distress. Although patients receiving palliative care have a broad range of underlying diagnoses and diverse end-of-life symptom trajectories [25,26,27,28], as found in the present study, poor control of pain and other distressing physical symptoms as death approaches affects a substantial proportion of patients.

There are several plausible interpretations of the finding in the community setting of reduced pre-death reporting of distressing symptoms by Indigenous compared with non-Indigenous patients. Possible (not mutually exclusive) explanations are: (i) selection bias, i.e., a lower threshold of clinical distress for admission to hospital rather than care in the community among Indigenous compared with other patients approaching death; (ii) ascertainment bias, for example due to greater stoicism or reluctance to report symptoms by Indigenous patients; (iii) other unmeasured confounding; (iv) chance findings among multiple comparisons; (v) Indigenous patients’ wellbeing and subjective experience of pain being particularly enhanced by care at home rather than in hospital [29]; or (vi) genuinely superior symptom management in the community at the very end of life for Indigenous compared with other patients.

Among the multiple comparisons between Indigenous and non-Indigenous patients in risks of reporting occurrence or worsening of distress related to individual symptoms, significant differences were few after correction for ‘false discovery’. These particular differences (in the symptoms of nausea and appetite problems) were not predicted a priori. It is notable nevertheless that the differences incorporating adjustment for age, sex, and diagnosis were uniformly in the direction of reduced risk among Indigenous patients. Distinguishing between genuinely lower symptom-induced distress among Indigenous patients and culturally modulated reticence in reporting of symptoms cannot be achieved in this analysis [30].

Our findings are strengthened by the large overall sample size and richly detailed information routinely recorded for PCOC on patients’ demographic and diagnostic characteristics, along with longitudinal assessments of clinical status. The study captures palliative care patients across a broad spectrum of demographic attributes and life-limiting illnesses. The one-decade average age difference between Indigenous and non-Indigenous palliative care patients is consistent with the well-recognised life-expectancy gap in Australia [31] (p. 235). However, this study has several limitations. Indigenous patients are substantially under-represented in PCOC data, accounting for approximately 1% of the total, a far lower proportion than the 2.8% of persons who identify as Indigenous in national census data [32]. Given the relatively small number of Indigenous patients, it was not feasible to explore differences in outcomes among diverse Indigenous populations, most importantly those who live in rural and remote areas and thereby face particular challenges in accessing culturally appropriate health care [33,34]. Although services accounting for more than 80% of specialist palliative care nationwide provided data to PCOC during the study period [16], these services are not representative of care provided to all Indigenous patients. Notably, no services from the Northern Territory, the jurisdiction with the highest proportion of Indigenous people, participated in PCOC during the study period. Furthermore, people living in rural and remote areas were known to be under-represented [18]. The patient-reported Symptom Assessment Scale has been validated for a general population [21] but, to our knowledge, not specifically for Indigenous Australian patients. Furthermore, Indigenous patients may have a higher threshold for divulging distress associated with symptoms such as pain and may be more reticent than others to report symptoms in general [15]. The irregularity of status ascertainment, and for many patients the paucity of clinical data points, precluded formal investigations of symptom trajectory such as group-based (‘latent class’) trajectory analysis [35]. Missing data may have biased the reported estimates, although missing values at the patient (‘trait’ rather than ‘state’) level resulted in the exclusion of less than 8% of records.

## 5. Conclusions

Indigenous patients cared for by specialist palliative care services nationwide that participate in PCOC do not appear to experience poorer control of symptoms in general than other patients. It cannot be determined from this study whether the lower risk of distress from symptoms reported by Indigenous compared with non-Indigenous patients receiving community-based care as death approaches reflects Indigenous patients experiencing genuinely lower levels of distress or that they are receiving better care that meets their needs. However, the findings provide reassurance that, in relation to symptom control, the quality of care provided to Indigenous people receiving palliative care from services registered with PCOC is as good as that provided to other Australians. There is room for further quantitative data linkage studies to delineate patient journeys throughout end-of-life illness more comprehensively, and complementary qualitative studies, to advance knowledge of end-of-life health care for Indigenous Australians.

## Figures and Tables

**Figure 1 ijerph-17-03131-f001:**
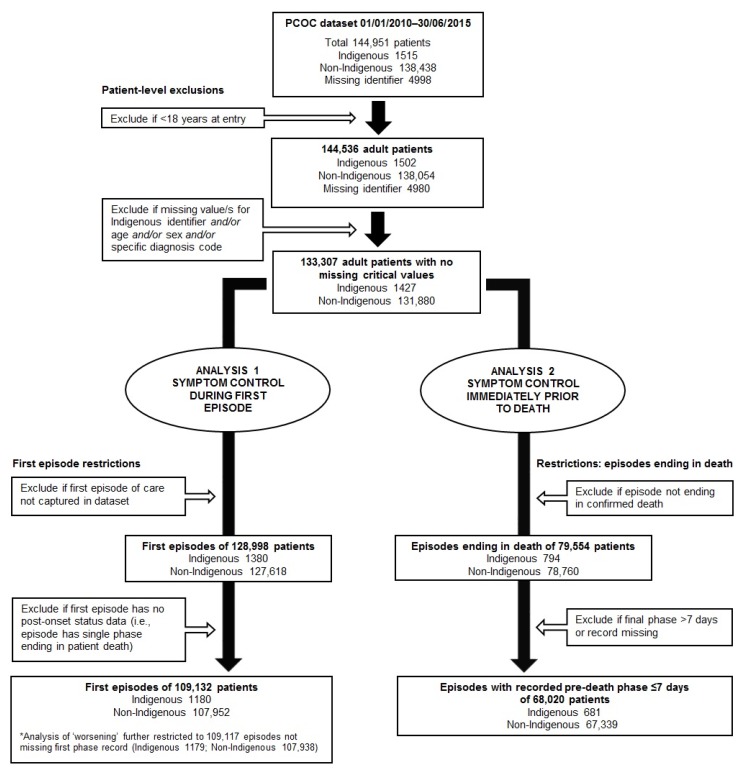
Scheme for inclusion in analyses of patients and their episodes.

**Table 1 ijerph-17-03131-t001:** Characteristics of patients and episodes.

Characteristic		Indigenous	Non-Indigenous
**First episode analysis**			
N		1180	107,952
Patient-level characteristics			
Age (years) at entry to care, mean (SD)		63.1 (14.3)	72.5 (13.5)
Age group at entry, n (%)			
	<65 years	625 (53.0)	28,168 (26.1)
	≥65 years	555 (47.0)	79,784 (73.9)
Sex, n (%)			
	Female	604 (51.2)	50,096 (46.4)
	Male	576 (48.8)	57,856 (53.6)
Principal Diagnosis, n (%)			
	Cancer	978 (82.9)	90,089 (83.5)
	Other	202 (17.1)	17,863 (16.5)
Episode/phase-level variables			
Setting of care, n (%)			
	Inpatient overnight admission	739 (62.6)	64,808 (60.0)
	Hospital ambulatory **	34 (2.9)	1755 (1.6)
	Community	407 (34.5)	41,389 (38.3)
Episode duration (days)		32.6	33.8
Mean		11	12
Median		1–717	1–1651
Range			
Number of phases in episode *		2.4	2.6
		2	2
		1–31	1–139
**Pre-death episode analysis**			
N		681	67,339
Patient-level characteristics			
Age (years) at entry to care, mean (SD)		64.2 (14.5)	73.9 (13.4)
Age group at entry, n (%)			
	<65 years	346 (50.8)	15,722 (23.3)
	≥65 years	335 (49.2)	51,617 (76.7)
Sex, n (%)			
	Female	345 (50.7)	30,848 (45.8)
	Male	336 (49.3)	36,491 (54.2)
Principal Diagnosis, n (%)			
	Cancer	527 (77.4)	52,204 (77.5)
	Other	154 (22.6)	15,135 (22.5)
Episode/phase-level variables			
Setting of care, n (%)			
	Inpatient overnight admission	561 (82.4)	54,206 (80.5)
	Hospital ambulatory **	0 (0.0)	53 (0.1)
	Community	120 (17.6)	13,080 (19.4)
Interval between symptom measurement and death (days)			
Mean		2.1	2.1
Median		1	1
Interquartile range		1–3	1–3
Correct recognition of phase as ‘terminal’ at phase start		521 (76.5)	51,022 (75.8)

* Excludes ‘bereavement’ phases, ** day admission or outpatient.

**Table 2 ijerph-17-03131-t002:** Patient-rated severity and worsening of symptom-related distress following commencement of a first episode of care: relative risks among Indigenous compared with non-Indigenous patients.

Model 1: All Settings, Crude												
Model 2: All Settings, Adjusted for Age + Sex + Principal Diagnosis												
Model 3: Subgroup Cared for in Inpatient Setting, Adjusted for Age + Sex + Principal Diagnosis												
Model 4: Subgroup Cared for in Community Setting, Adjusted for Age + Sex + Principal Diagnosis												
Symptom(s)	Model 1			Model 2			Model 3			Model 4		
	RR	*p*	RR (95% CI)	RR	*p*	RR (95% CI)	RR	*p*	RR (95% CI)	RR	*p*	RR (95% CI)
**Occurrence of moderate-to-severe (SAS score 4–10) symptom/s following commencement of an episode**												
Any symptom	0.99	0.682	(0.96–1.03)	0.97	0.115	(0.94–1.01)	0.99	0.792	(0.95–1.04)	0.96	0.084	(0.91–1.01)
≥3 symptoms	0.96	0.313	(0.89–1.04)	0.91	0.013	(0.84–0.98)	0.94	0.211	(0.85–1.04)	0.89	0.060	(0.79–1.01)
Pain	1.07	0.087	(0.99–1.16)	0.97	0.403	(0.89–1.05)	0.97	0.508	(0.87–1.07)	0.97	0.660	(0.85–1.11)
Nausea	0.90	0.231	(0.76–1.07)	0.79	0.005 *	(0.67–0.93)	0.82	0.071	(0.66–1.02)	0.80	0.091	(0.61–1.04)
Breathing problem	1.02	0.624	(0.93–1.13)	1.00	0.925	(0.91–1.10)	1.00	0.972	(0.89–1.13)	1.03	0.732	(0.88–1.20)
Bowel problems	0.90	0.080	(0.79–1.01)	0.88	0.033	(0.78–0.99)	0.89	0.099	(0.77–1.02)	0.83	0.119	(0.66–1.05)
Appetite problems	0.88	0.007	(0.81–0.97)	0.87	0.003 *	(0.80–0.96)	0.93	0.233	(0.83–1.05)	0.84	0.017	(0.74–0.97)
Insomnia	1.08	0.155	(0.97–1.21)	0.97	0.596	(0.87–1.08)	1.06	0.436	(0.92–1.22)	0.92	0.355	(0.76–1.10)
Fatigue	0.97	0.202	(0.92–1.02)	0.95	0.037	(0.90–1.00)	0.99	0.814	(0.93–1.06)	0.91	0.013	(0.85–0.98)
**Worsening (SAS score increment ≥ 2) of symptom/s at any point following commencement of an episode**												
Any symptom	0.95	0.111	(0.90–1.01)	0.95	0.069	(0.89–1.00)	0.96	0.289	(0.88–1.04)	0.95	0.244	(0.88–1.03)
≥3 symptoms	0.97	0.604	(0.86–1.09)	0.94	0.319	(0.84–1.06)	0.98	0.854	(0.83–1.16)	0.99	0.859	(0.84–1.15)
Pain	0.97	0.655	(0.87–1.09)	0.95	0.380	(0.85–1.07)	0.96	0.618	(0.82–1.13)	0.99	0.930	(0.85–1.16)
Nausea	0.85	0.079	(0.70–1.02)	0.77	0.006*	(0.64–0.93)	0.75	0.051	(0.57–1.00)	0.86	0.224	(0.67–1.10)
Breathing problems	0.95	0.412	(0.83–1.08)	0.94	0.336	(0.82–1.07)	0.96	0.665	(0.8–1.15)	0.96	0.706	(0.78–1.18)
Bowel problems	0.94	0.398	(0.82–1.08)	0.94	0.411	(0.82–1.08)	0.95	0.535	(0.79–1.13)	0.97	0.795	(0.78–1.21)
Appetite problems	0.94	0.347	(0.83–1.07)	0.94	0.303	(0.82–1.06)	1.04	0.657	(0.87–1.24)	0.91	0.304	(0.76–1.09)
Insomnia	1.05	0.510	(0.91–1.20)	1.00	0.984	(0.87–1.15)	1.11	0.291	(0.92–1.34)	0.96	0.665	(0.78–1.17)
Fatigue	0.94	0.226	(0.84–1.04)	0.93	0.204	(0.84–1.04)	0.91	0.196	(0.78–1.05)	1.01	0.932	(0.87–1.17)

CI: confidence interval, RR: Relative risk, SAS: Symptom Assessment Scale. * Significant after correction for multiple comparisons (Benjamini–Hochberg method [23]).

**Table 3 ijerph-17-03131-t003:** Relative risks of moderate-to-severe symptom-related distress at final assessment prior to death: Indigenous compared with non-Indigenous patients.

Model 1: All Settings, Crude												
Model 2: All Settings, Adjusted for Age + Sex + Principal Diagnosis + Interval between Assessment and Death (i.e., Final Phase Length)												
Model 3: Subgroup Cared for in Inpatient Setting, Adjusted for Age + Sex + Principal Diagnosis + Interval between Assessment and Death												
Model 4: Subgroup Cared for in Community Setting, Adjusted for Age + Sex + Principal Diagnosis + Interval between Assessment and Death												
Symptom(s)	Model 1			Model 2			Model 3			Model 4		
	RR	*p*	RR (95% CI)	RR	*p*	RR (95% CI)	RR	*p*	RR (95% CI)	RR	*p*	RR (95% CI)
Any symptom	1.00	0.870	(0.94–1.05)	0.97	0.400	(0.91–1.04)	1.03	0.362	(0.96–1.11)	0.78	0.001 *	(0.67–0.91)
≥3 symptoms	1.04	0.571	(0.92–1.17)	0.93	0.333	(0.81–1.07)	1.01	0.898	(0.87–1.17)	0.65	0.030	(0.45–0.96)
Pain	1.12	0.051	(1.00–1.25)	0.98	0.743	(0.87–1.11)	1.04	0.545	(0.91–1.19)	0.68	0.044	(0.47–0.99)
Nausea	0.83	0.195	(0.62–1.10)	0.72	0.049	(0.52–1.00)	0.77	0.137	(0.54–1.09)	0.53	0.150	(0.22–1.26)
Breathing problem	1.09	0.105	(0.98–1.22)	1.02	0.741	(0.91–1.14)	1.07	0.246	(0.95–1.21)	0.79	0.180	(0.57–1.11)
Bowel problems	0.84	0.065	(0.69–1.01)	0.79	0.035	(0.64–0.98)	0.81	0.076	(0.65–1.02)	0.69	0.213	(0.38–1.24)
Appetite problems	0.82	0.010	(0.71–0.95)	0.81	0.013	(0.68–0.96)	0.90	0.255	(0.74–1.08)	0.55	0.004 *	(0.37–0.83)
Insomnia	1.11	0.256	(0.92–1.34)	0.97	0.806	(0.79–1.21)	0.96	0.777	(0.75–1.24)	1.02	0.910	(0.68–1.55)
Fatigue	0.97	0.426	(0.89–1.05)	0.95	0.265	(0.86–1.04)	1.00	0.984	(0.90–1.11)	0.80	0.029	(0.66–0.98)

CI: confidence interval, RR: relative risk; * Significant after correction for multiple comparisons (Benjamini–Hochberg method [23]).

## Data Availability

Restrictions apply to the availability of the data that support the findings of this study, which were used under license, and so are not publicly available. Under the User Agreement for Release of PCOC Data, the authors have undertaken (21 June 2015) ‘not to disclose the Confidential Information [i.e., de-identified individual patient records] in any released output’. Non-identifiable extracts of data from the PCOC longitudinal database can be made available for use in research. Researchers must apply in writing and provide detailed project descriptions for approval by the PCOC Executive Directors Group: https://ahsri.uow.edu.au/pcoc/4researchers/accessingdata/index.html. The authors of the current study accessed the data by this means. Details of variables included in the PCOC dataset are available online in the PCOC Data Dictionary & Technical Guidelines: http://tinyurl.com/PCOC-V3-DataDictionary.

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
