# Peer review of "Symptom-Related Distress among Indigenous Australians in Specialist End-of-Life Care: Findings from the Multi-Jurisdictional Palliative Care Outcomes Collaboration Data"

_ijerph, 2020, doi:10.3390/ijerph17093131_

Round 1
Reviewer 1 Report
The manuscript is well-written and the conclusions and their significance are very clear. This manuscript deserves publication with formatting and, either, more basic models included in the supplementary materials or the code/data released (or both, with luck!). Also, the manuscript would benefit from writing out the equations for the models.
However, there are some statistical issues. The authors mention multiple comparisons but do not correct for them. Correct for the FWER, FDR, or both, and justify how you do it (multiple ways are better; even if they change the result, you can still talk about your result as-is with caveats). The justification that symptom-specific differences were in the same direction with covariates is not convincing. In point of fact, these sorts of controls can induce confounding (think Judea Pearl and Donald Rubin) and inflate variance (include VIFs). In fact, just running all of the standard econometric robustness checks and, in the discussion/limitations talking about how to achieve d-separation, would be good. The fact that some talked-about results were only significant in the most parameterized models is alarming and, while it doesn't mean they can't be discussed, it should be mentioned that the results are exploratory in nature.
Because of the near-significance of the differences, a large overall sample size weakens the conclusions. The reason for this is Lindley's Paradox, or that as sample size increases, a greater proportion of hypotheses become 'significant'. In order to account for it, please scale the p values accordingly. If you don't know how to do this, consult Good (1988) or Naaman (2016). A large sample size - since (with assumptions) it decreases sampling error - only increases the convincingness of a result with this in mind.
Normally I would not say to accept a paper that does not release its data, but that is both not a policy of this journal and understandably not possible for this data - do files should still be released -, so with some minor alterations, I would urge acceptance.
Good, I. J. (1988). The Interface Between Statistics and Philosophy of Science. Statistical Science, 3(4), 386–397. https://doi.org/10.1214/ss/1177012754
Naaman, M. (2016). Almost sure hypothesis testing and a resolution of the Jeffreys-Lindley paradox. Electronic Journal of Statistics, 10(1), 1526–1550. https://doi.org/10.1214/16-EJS1146Author Response
Please see the attachment.

Reviewer 2 Report
This is an interesting paper
For non-Australians it may be of interest to show the proportion of Indigenous: Non-indigenous in the overall population. In the study it is 1%: 99% and it would be interesting to know how this compares to the population.
Reviewer 3 Report
Thank you for the opportunity to review the manuscript ijerph-780746 entitled “Symptom-related distress among Indigenous Australians in specialist end-of-life care: findings from the multijurisdictional Palliative Care Outcomes Collaboration data”.
In this retrospective dataset analysis, the authors provide additional perspectives to their previously published analysis of the same dataset in PLoS ONE 2019, 14 comparing symptom-related distress among Indigenous and non-Indigenous patients in specialized palliative care. This article analyses data that must be incomplete when dealing with a population so different in cultural and psychological aspects – all of which are important in palliative care.
Overall I think this paper is suitable for publication in the IJERPH.
Summary:
The authors compare symptom-related distress among Indigenous and non-Indigenous patients at beginning of care and prior to death using a large dataset spreading over 66 months. After elaborating on the inclusion criteria the authors use the Symptom Assessment Scale as their tool to measure symptom-related distress. They model their data on different care settings as inpatient care and community care. Over all, the results show no striking differences between the two groups of patients. However, Indigenous patients had a lower risk of reporting nausea, bowel and appetite problems as well as fatigue, and three or more symptoms especially in a community setting. This was found in the first episodes analyzed as well as in the pre-death episodes. The authors discuss their findings and if they can stand for a quality indicator in this special population, they address weaknesses and unclear aspects of their study appropriately and in detail and set out further research strategies to address them.
These are my minor remarks in detail:
- Introduction:
- The authors lead into the subject giving an overview on problems of Indigenous patients
- Then, they introduce their study concept as part of a quality assessment in palliative care
- In my opinion the introduction needs no modification.
- Materials and Methods:
- 1.: Description of data source. In line 94/95 the overlap of the datasets is briefly mentioned, for me, mentioning the amount of overlap would add clarity (it should be around 15% according to the flowchart)
- 2.: Inclusion criteria. The flowchart nicely clarifies the strategy – no modification needed.
- 3.: Raw data acquisition. Clearly written, for the sake of clarity one could add that the SAS scoring system is not validated in Indigenous peoples. Otherwise this paragraph needs no modification.
- Results:
- Line 167: This difference in age has been found in their previous analysis as well so the similarity should be mentioned.
- 1. and 3.2.: No modification needed
- Discussion:
- In the discussion, the authors try several interpretations stressing the natural lack of cultural and psychological factors in their data as well as missing data from a territory highly populated by Indigenous people.
- In my opinion this paragraph needs no revision.
Reviewer 4 Report
The paper is original and interesting for other lands in which coexist different cultures
